# Resolving Multi-Path Interference in Compressive Time-of-Flight Depth Imaging with a Multi-Tap Macro-Pixel Computational CMOS Image Sensor

**DOI:** 10.3390/s22072442

**Published:** 2022-03-22

**Authors:** Masaya Horio, Yu Feng, Tomoya Kokado, Taishi Takasawa, Keita Yasutomi, Shoji Kawahito, Takashi Komuro, Hajime Nagahara, Keiichiro Kagawa

**Affiliations:** 1Graduate School of Integrated Science and Technology, Shizuoka University, Shizuoka 422-8011, Japan; mhori@idl.rie.shizuoka.ac.jp (M.H.); feng.yu.16@shizuoka.ac.jp (Y.F.); kokado.tomoya@mail.canon (T.K.); 2Research Institute of Electronics, Shizuoka University, Shizuoka 422-8011, Japan; ttakasawa@idl.rie.shizuoka.ac.jp (T.T.); kyasu@idl.rie.shizuoka.ac.jp (K.Y.); kawahito@idl.rie.shizuoka.ac.jp (S.K.); 3Graduate School of Science and Engineering, Saitama University, Saitama 338-8570, Japan; komuro@mail.saitama-u.ac.jp; 4Institute of Datability Science, Osaka University, Osaka 565-0871, Japan; nagahara@ids.osaka-u.ac.jp

**Keywords:** time-of-flight depth imaging, multi-path interference, multi-tap macro-pixel, charge modulators, compressive sensing, CMOS image sensor

## Abstract

Multi-path interference causes depth errors in indirect time-of-flight (ToF) cameras. In this paper, resolving multi-path interference caused by surface reflections using a multi-tap macro-pixel computational CMOS image sensor is demonstrated. The imaging area is implemented by an array of macro-pixels composed of four subpixels embodied by a four-tap lateral electric field charge modulator (LEFM). This sensor can simultaneously acquire 16 images for different temporal shutters. This method can reproduce more than 16 images based on compressive sensing with multi-frequency shutters and sub-clock shifting. In simulations, an object was placed 16 m away from the sensor, and the depth of an interference object was varied from 1 to 32 m in 1 m steps. The two reflections were separated in two stages: coarse estimation based on a compressive sensing solver and refinement by a nonlinear search to investigate the potential of our sensor. Relative standard deviation (precision) and relative mean error (accuracy) were evaluated under the influence of photon shot noise. The proposed method was verified using a prototype multi-tap macro-pixel computational CMOS image sensor in single-path and dual-path situations. In the experiment, an acrylic plate was placed 1 m or 2 m and a mirror 9.3 m from the sensor.

## 1. Introduction

Time-of-flight (ToF) [1] depth imaging is a technique for measuring the depth between a camera and objects based on the round-trip time of light emitted from the camera, assuming that the speed of light is constant. ToF has been applied to a variety of applications [2] such as autonomous driving, robot navigation, modeling of objects, and gesture recognition in entertainment due to the small device size, real-time measurement, and the ability to measure texture-less surfaces. However, ToF has a disadvantage in that the measurement accuracy is degraded by multi-path interference [3,4]. There are four major types of multi-path interference: (1) surface reflections from transparent objects, (2) multiple reflections between objects, (3) sub-surface scattering, and (4) volumetric scattering caused by bulky weak scattering media such as smoke or fog. In this paper, we focus on surface reflections.

There are two types of ToF image sensors: direct (dToF) [5] and indirect (iToF) [6]. In dToF, impulse light with a duration of nanoseconds is mostly used, and the temporal reflected light waveform is measured pixel by pixel by using a single-photon avalanche diode (SPAD) [7]. Comparison of dToF and iToF is summarized in Table 1. In iToF, on the other hand, the inner products or correlations of the reflected light waveform with multiple time-window functions are measured. Then, the temporal or phase delay is calculated. To perform the inner-product operation, charge modulators are utilized. iToF is further classified into amplitude modulation continuous wave (AMCW) iToF [8] and pulse iToF [9]. AMCW iToF uses sinusoidally modulated light, and pulse iToF uses rectangular pulsed light with a duration of tens of nanoseconds. dToF provides the genuine waveform of the reflected light for many temporal sampling points (typically hundreds of points or more) with a high temporal resolution by using high-precision time-to-digital converters (TDCs). Although separation of multi-path components is easier by dToF than by iToF because the whole waveform of the reflected light is obtained, large digital circuits are required to build the histograms of photon arrival times. iToF is advantageous for achieving a small sensor size because signal detection is performed in the charge domain in pixels, and large digital or analog circuits are required for detection. However, the number of temporal or phase sampling points of the detected signal is only a few, typically three or four. Therefore, multi-path component separation is more difficult in iToF than in dToF.

To decompose or compensate for the surface-reflection-type multi-path interference, methods that use multiple modulation frequencies or delays have been demonstrated [10,11,12,13,14]. However, these methods are vulnerable to motion artifacts, since multiple images with some scanning are necessary. Recently, deep-learning-based methods have been emerging [15,16]. Reference [16] proposed Deep ToF, which uses a single depth image taken by an ordinary AMCW iToF camera. Then, the multi-path interference is corrected by deep learning in real time. This method uses information about neighboring pixels or the scene to correct multi-path interference caused by multiple reflections. However, it could be difficult to resolve multi-path interference when neighboring pixels do not provide any cue to correct the multi-path interference. This problem could arise for the case of surface-reflection-type multi-path interference. To achieve motion-artifact-free single-shot imaging acquisition in ToF, specially designed image sensors are necessary.

In our previous study [17,18], we have proposed a method for multi-path component separation using multi-aperture-based temporally compressive pulse iToF. This method requires no large on-chip digital circuits, and the number of temporal sampling points is much more than that of conventional iToF, by taking advantage of the sparsity of the ToF signal. Figure 1 shows a method for separating multi-path components based on compressive sensing [19,20,21,22]. In this method, 15 inner products of the reflected pulse light with binary random shutters are acquired at the same time. Then, 32-bin reflected light histograms are reproduced. Thus, the drawback of iToF, i.e., the limited number of sampling points, is alleviated, and multiple reflected light paths are separated. However, this method requires a custom lens array, so it is not easy to reconfigure the optics. In addition, it is necessary to compensate for the parallax among the lenses. In the framework in [18], the depth resolution is the same as the width of the minimum time window, although sub-window resolution is achieved in ordinary iToF cameras.

We have proposed a multi-tap macro-pixel-based compressive ultra-high-speed CMOS image sensor [23]. This image sensor can implement multi-frequency shutters and sub-clock shifting. Such features are useful for multi-path component separation in a single shot. By using a macro-pixel structure instead of a multi-aperture structure, it is possible to capture images using ordinary single-aperture lenses so that there is no disparity problem. In this paper, we evaluate the potential performance of separating dual-path components based on the multi-tap macro-pixel CMOS image sensor with multiple modulation frequencies and sub-clock shifting by simulation. Then, we demonstrate dual-path component separation in temporally compressive pulse iToF with a prototype image sensor. The depth estimation is performed in two steps: (1) separation of multiple peaks using a compressive sensing solver TVAL3 [24,25] and (2) refining the depths with sub-clock accuracy by fitting the result obtained at step 1 by a nonlinear optimization method [26]. Step 2 enables sub-clock-resolution depth estimation, which was not achieved in our previous study. Although this method is not suitable for real-time processing, the purpose of this study is to investigate the potential performance of the proposed scheme.

In Section 2, the multi-tap macro-pixel compressive image sensor architecture and depth imaging based on the temporally compressive pulse iToF are described. Section 3 shows system modeling and performance evaluation by simulation. In Section 4, dual-path component separation is demonstrated. In the experiment, a transparent acrylic plate was placed between the camera and an objective mirror to introduce multi-path interference. Section 5 discusses the issues of the proposed method. Section 6 concludes this paper.

## 2. Temporally Compressive Time-of-Flight Depth Imaging

### 2.1. Multi-Tap Macro-Pixel Computational CMOS Image Sensor

Figure 2 shows the architecture of a multi-tap macro-pixel computational CMOS image sensor. The sensor is mainly composed of a shutter controller, a clock tree, a lateral electric field charge modulator (LEFM) [27,28] driver, an array of macro-pixels, and readout circuits (RDOs). The specifications are summarized in Table 2.

The sensor was fabricated in a 0.11 µm CMOS image sensor process. A macro-pixel is composed of 2 × 2 subpixels. Each subpixel is implemented by a four-tap LEFM, which has four storages (FD1-4). Charge transfer to the FDs and charge drain is controlled by five gate signals (G1-4 and GD). The shutter pattern of each gate is programmable. The shutter length per tap can be set to 8–256 bits in steps of 8 bits. Here, the shutter length is defined as the length of binary shutters and represented by bits. In Figure 1b, the shutter code in a unit time is given by one bit.

This sensor compresses images when the shutter length is longer than 16 bits (=the number of the total taps in the macro-pixel or the number of measured signals per macro-pixel). The mathematical representation of compressive sensing is briefly mentioned below. When we consider an N-dimensional column vector x, an M×N matrix A, and an M-dimensional column vector y, the relationship is denoted by
(1)y=Ax.

Note that x is the original input signal, and y is the signal measured through a known measurement matrix A. When N>M, the original signal x is compressed. We can reproduce the original input signal x by solving the inverse problem from y and A based on a sparsity constraint.

Figure 3 shows the imaging process of a multi-tap macro-pixel computational CMOS image sensor. First, each tap detects the optical signal from the objects using different temporal shutters. Then, different temporally compressed images are obtained. After solving the inverse problem from the temporally compressed images and shutter patterns based on the sparsity, the time-sequential images are reproduced.

For simplicity, the image acquisition and reproduction are explained based on a single pixel; i.e., x is a temporal waveform of the received light for a single pixel. The coded shutter is represented by the measurement matrix A in compressive sensing. It is assumed that A is composed of M sets of temporal shutters given by the M×N-dimensional matrix. Then, Equation (1) is rewritten as
(2)$$\left(\begin{array}{c} \begin{array}{c} y_{1}\\ y_{2}\\ \vdots \end{array}\\ y_{M} \end{array}\right)=\left(\begin{array}{ccc} a_{11} & \cdots & a_{1N}\\ \vdots & \ddots & \vdots \\ a_{M1} & \cdots & a_{MN} \end{array}\right)\left(\begin{array}{c} \begin{array}{c} x_{1}\\ x_{2}\\ \vdots \end{array}\\ x_{N} \end{array}\right),$$
where each row of A describes one temporal sequence of a shutter. The total number of taps is the same as the number of taps in a subpixel times the number of subpixels in a macro-pixel, namely M = 4 × 4 = 16. If the original input signal x is K-sparse, which means only K elements have non-zero values and all the other elements are zero, x is estimated by the following optimization based on the $l_{0}$-norm:(3)$${\hat{x}}^{\left(0\right)}=\underset{x}{\mathrm{argmin}}∥x{∥}_{0}\;\mathrm{subject}\;\mathrm{to}\;y=Ax.$$

Here, $∥x{∥}_{m}$ shows $l_{m}$-norm of x. Note that $l_{0}$-norm is defined as the number of the non-zero elements in x. However, solving this problem is hard because this is a huge combinatorial problem. Therefore, $l_{1}$-norm minimization is used instead, as follows:(4)$${\hat{x}}^{\left(1\right)}=\underset{x}{\mathrm{argmin}}∥x{∥}_{1}\;\mathrm{subject}\;\mathrm{to}\;y=Ax.$$

Total variation (TV) minimization is widely used to solve $l_{1}$-norm minimization, where the original input signal x is estimated by
(5)$${\hat{x}}^{\left(TV\right)}=\underset{x}{\mathrm{argmin}}\Sigma_{i}∥D_{i}x{∥}_{1}\;\mathrm{subject}\;\mathrm{to}\;y=Ax.$$

Here, $D_{i}$ is the differential operator that subtracts an adjacent element from the i-th element.

### 2.2. Modeling of Multi-Path Interference in ToF

The detected light waveform g(t) at the ToF camera, shown in Figure 4c, is modeled as follows:(6)g(t)=L(t)∗f(t),
where f(t) (Figure 4a) is the scene response function, L(t) (Figure 4b) is the waveform of the total system response, and ∗ is the convolution operator. In the multi-path scenario, f(t) is modeled as
(7)$$f\left(t\right)=\overset{K}{\underset{i=1}{\sum }}a_{i}\delta \left(t-\frac{2d_{i}}{c}\right),$$
where $a_{i}$ is the amplitude and $d_{i}$ is the depth of the i-th reflection, c is the speed of light, K is the total number of reflections, and δ(x) is the δ-function. L(t) is represented by the convolution of the light source waveform $L_{0}\left(t\right)$ (Figure 4d) and the sensor response $h_{s}\left(t\right)$ (Figure 4e):(8)$$L\left(t\right)=L_{0}\left(t\right)*h_{s}\left(t\right).$$

The sensor response $h_{s}\left(t\right)$ is further represented by the convolution of the response of the charge modulator $h_{s1}\left(t\right)$ and the photodiode response $h_{s2}\left(t\right)$. When either of these responses is dominant, $h_{s}\left(t\right)$ can be approximated by the first-order delay:(9)$$h_{s}\left(t\right)=h_{s1}\left(t\right)*h_{s2}\left(t\right)\sim \exp\left(-\frac{t}{\tau }\right),$$
where τ is the time constant of the modulator or photodiode. Thus, the detected light waveform g(t) at the ToF camera is shown in Figure 4c and is modeled as
(10)$$g\left(t\right)=L\left(t\right)*f\left(t\right)\sim L_{0}\left(t\right)*\exp\left(-\frac{t}{\tau }\right)*\overset{K}{\underset{i=1}{\sum }}a_{i}\delta \left(t-\frac{2d_{i}}{c}\right).$$

The image sensor puts out *M* compressed signals $\{p_{m}\}\;\left(m=1,\;\ldots ,\;M\right)$ (Figure 4f), which is also denoted by a vector representation, $p={\left(p_{1},\ldots ,\;p_{M}\right)}^{T}$. Note that $x^{T}$ means the transpose of a vector, x. When the shutter pattern for the m-th tap is written as $w_{m}\left(t\right)$, the detected signal $p_{m}$ is given by the inner product or correlation between $w_{m}\left(t\right)$ and g(t):(11)$$p_{m}=\int w_{m}\left(t\right)g\left(t\right)dt.$$

### 2.3. Selection of Exposure Patterns

As mentioned in Section 2.1, 16 different shutters are applied to the multi-tap macro-pixel computational CMOS image sensor. In this paper, 32-bit shutter patterns shown in Figure 5 are used. The shutter patterns of subpixel 1 are shifted by half a clock from those of subpixel 2. Time windows for slower frequencies can increase the depth range, while high-frequency time windows improve the depth resolution. By using half-clock shifting, the depth resolution can be improved by a factor of two without shortening the width of the minimum time window or increasing the operating frequency of the sensor.

When ordinary four-tap pulse iToF images sensors [29] are used for the dual-path scene, the shutter patterns only for one frequency can be applied at once, e.g., the shutter patterns for subpixel 4 in Figure 5. They cannot resolve the multi-path interference if two reflections are detected in the same time windows. It is possible for them to emulate the shutter patterns in Figure 5. However, image acquisition should be performed four times for each set of the shutter patterns for each subpixel. Thus, this implementation will suffer from motion artifacts. The benefit of our multi-tap macro-pixel computational CMOS image sensor is single-shot image acquisition. A drawback is that the macro-pixel is larger in size than ordinary four-tap pixels.

### 2.4. Solving the Inverse Problem and Depth Refinement

The depth estimation process is pixelwise and composed of two stages. We assume that there are two reflections: objective light and interference light. This algorithm can be easily extended to more reflections.

First, we reconstruct the scene response function denoted by $\hat{x}$ from the sensor output p**,** which is a measured version of y, and the measurement matrix A. x is a discrete version of f(t), and A describes the shutter patterns $\left\{w_{m}\left(t\right)\right\}$ and the total system response L(t). Next, we find peaks in $\hat{x}$ and obtain *K* peak depths $\left\{d_{i}\right\}$ and their amplitudes $\left\{a_{i}\right\}$ (i=1, …, K). When two adjacent peaks are too close, only one peak is found. To overcome this problem, we add two peaks at both ends of the highest peak that is most likely to include the objective light. Their amplitudes are half of the peak. Then, for all combinations of two arbitrary peaks among the *K* + 2 peaks, their depths and amplitudes are refined by a nonlinear search algorithm. Refinement is performed by minimizing the following evaluation function for two peaks *i* and *j* (*i*, *j* = 1, …, K+2, i≠j):(12)E(a=ajai, di, dj)=|F(a, di, dj)−p∑m=1Mpm|2.

Here, the amplitudes of the sensor output are normalized. F is the forward problem function and provides a simulated sensor output for $a,\;d_{i},\;\mathrm{and}\;d_{j}$, where a is the ratio of the two amplitudes. The sensor output is normalized by the sum of the elements, i.e., (y1, …, yM)T∑m=1Mym. The set of the two peaks that gives the minimal evaluation value is selected as a solution.

## 3. Simulation

### 3.1. Simulation Method

The simulation flow is shown in Figure 6. For simplicity, only photon shot noise is considered, whereas sensor random noise is neglected. The shot noise is given as follows: First, we determine the total number of photons for the *M* taps, which is given by $N_{\mathrm{op}}$, to define the intensity of the reflected light or signal level. The sensor output p is scaled to make $\overset{M}{\underset{m=1}{\sum }}p_{m}$ equal to $N_{\mathrm{op}}$. Then, shot noise is given to each scaled $p_{m}$ based on the Poisson distribution to generate $p^{\prime }$ electrons. The depth and amplitude are estimated from $p^{\prime }$ by the method explained above. This process is repeated *R* times to determine the precision and accuracy of the estimated depths, which are evaluated by using the relative standard deviation (RSD) and relative mean error (or simply error). Note that RSD and error refer to precision (or uncertainty) and accuracy (or nonlinearity), respectively. We compare these indexes for different $N_{\mathrm{op}}$’s and a’s.

As for the multi-path interference, single-path and dual-path scenarios are simulated. The single-path situation is essential in the depth estimation. It is necessary to test if the proposed algorithm correctly works for this simplest case. The resolution of multi-path interference in the dual-path situation is the main issue in this study. It is expected that the estimated depth of the objective light becomes erroneous as the number of photons decreases or the amplitude of the interference light increases. With the depth of the objective light fixed, the dependency of the estimated depths for both objective and interference light on the depth of the interference light is investigated.

For the number of photons, two different cases are discussed. In reality, the number of the received photons at the ToF image sensor decays in inverse proportion to the square of the depth of the object. In the simplest case, this decay is not considered to investigate the natural behavior of the proposed algorithm. Then, the decay of the received photon number is considered.

The simulation conditions were as follows: The clock frequency for the shutter generator was 73 MHz. Therefore, the unit time was 13.7 ns which corresponded to approximately 2 m in depth. The measurable depth range was 32.64 m for the shutter length of 32 bits. The time step in the simulations was 0.05 ns (≈0.75 cm in depth). The time constant τ of the system response in Equation (9) was set to 1 ns. The duration of the laser pulse was as long as the shortest time window, namely 13.7 ns. *R* was set to 100.

To generate the shot noise based on the Poisson distribution, the MATLAB function imnoise was used. TVAL3 was used as a compressive sensing solver [24,25]. The parameters of TVAL3 were as follows: nonneg was true, μ was 2^4^, β was 2^−2^, $\mu_{0}$ was 2^1^, $\beta_{0}$ was 2^−3^, and other parameters were the default values. To find peaks, findpeaks in MATLAB Signal Processing Toolbox was used. To refine the peak positions, fminsearchbnd [26] was used to apply non-negative constraints.

### 3.2. Single Path

First, we simulated single paths (with no interference light) for the following conditions: $N_{\mathrm{op}}$ was 5000, 10,000, 20,000, or 40,000. The amplitude ratio a was 1. Depth-dependent decay of the number of photons was not considered. The depth d was varied from 1 to 32 m in steps of 1 m. The simulation results in Figure 7 show that depth errors were very small over the range. It was confirmed that the proposed method worked normally even for the single-path scenario.

Then, simulation was conducted for a realistic case considering the depth-dependent decay of light intensity. Figure 8 shows the simulation results. As $N_{\mathrm{op}}$ decreased, RSD increased because the shot noise relatively increased. The waveforms of RSD were oscillating a little, but mostly constant. No nonlinearity was observed, as shown in Figure 8b.

### 3.3. Dual-Path

In the dual-path scenario, two situations were considered. In Type A, the total photon number was changed while the amplitude was fixed. In Type B, the amplitude was changed with the total photon number fixed. All the conditions are shown in Table 3. $d_{1}$ and $a_{1}$ are for the objective light. $d_{2}$ and $a_{2}$ are for the interference light. The reflection’s depth $d_{1}$ and amplitude $a_{1}$ were always 16 m and 1, respectively. The interference depth $d_{2}$ was moved from 1 m to 32 m in 1 m steps. Because this is a numerical investigation, the interference light can come after the objective light.

Firstly, simulations without the depth-dependent decay of the number of photons were performed. Figure 9 shows the results of the simulation. Both the relative error and RSD of $d_{1}$ (objective reflection) became large as $N_{\mathrm{op}}\;\mathrm{decreased}\;\mathrm{or}\;a_{2}\;\mathrm{increased},\;\mathrm{especially}\;\mathrm{when}$ the objective light and interference light were merged. As shown in the results of Type A (Figure 9a–d), the absolute value of the relative mean error for different light intensities was <1.2% in both $d_{1}$ and $d_{2}$ for all given $N_{\mathrm{op}}$’s except when $d_{2}$ was around 1 m or 16 m. The results of Type B are shown in Figure 9e–h. The absolute value of the relative mean error for different amplitudes of the interference light was <0.8% in both $d_{1}$ and $d_{2}$ except when $d_{2}$ was around 1 m, 16 m, or 26 m. The reason why the depth error at around 1 m is relatively large is that the relative errors become large as the distance becomes small. Other issues related to the large errors are discussed below.

Figure 10 shows the simulation results for Type B when the depth-dependent decay of photons was considered. $N_{\mathrm{op}}$ was adjusted to be 5000 when the depth was 4 m and the amplitude was 1, and it was in inverse proportion to the square of depth. RSD and error of $d_{1}$ became large as the contribution of the interference light increased. The behavior of RSD and error of $d_{2}$ is interesting. As the amplitude of the interference light becomes small, both RSD and error become large. At far distances, the nonlinearity of $d_{2}$ for $a_{2}=0.1$ is significantly large. This is possibly because $d_{2}$ becomes closer to $d_{1}$ = 16 m.

## 4. Experiment Using a Multi-Tap Macro-Pixel Computational CMOS Image Sensor

### 4.1. Experimental System

In this section, the aforementioned method is demonstrated with a prototype multi-tap macro-pixel computational CMOS image sensor shown in Table 2 and Figure 2. The experimental system setup is shown in Figure 11. A semiconductor pulsed laser (Tama Electric, Hamamatsu, Japan, Model LDS-320, λ=850 nm) emitted light with a duration of 21 ns toward the objective mirror and the acrylic plate through a mirror having a hole in the middle (holed mirror) to realize a semi-coaxial configuration. The distance between the acrylic plate and the holed mirror was 1 or 2 m. The objective mirror was placed 9.3 m away from the holed mirror. The 16 compressed images for the 16 32-bit shutters shown in Figure 5 were acquired in a single shot. The shutter controller frequency was set to 73 MHz. In this experiment, no imaging lens was used. The objective and interference light directly illuminated the image sensor. In solving the inverse problem, it was assumed that the subpixels in the same macro-pixel were uniformly illuminated. The saturation level of the sensor was 67558 LSB. The read noise of the sensor was 376 LSB.

The correlation of the total system response of the ToF camera system and the coded shutters, which was equivalent to the sensor output p, was measured before the ToF imaging. The measurement was performed while the emission timing delay of the pulsed laser t was scanned over the measurable range. The measured functions are denoted as follows:(13)$${p^{\prime }}_{m}\left(t\right)=\int w_{m}\left(\xi \right)L\left(\xi -t\right)d\xi .$$

The measured responses shown in Figure 10 were used in every step of the inverse problem solving. The measurement matrix A was generated by down-sampling $\left\{{p^{\prime }}_{m}\left(t\right)\right\}$, and the sensor output in the refinement stage was given by the superposition of the sensor output in Figure 12 for different times of flight (=light pulse delays) such as
(14)$$p_{m}=\int {p^{\prime }}_{m}\left(t\right)f\left(t\right)dt.$$

### 4.2. Measured and Processed Results

Figure 13 shows 16 temporally compressed images when the objective mirror and the acrylic plate were placed at 9.3 m and 1 m, respectively. For comparison, single-path scenes were also measured. Figure 13a,b show the captured images for the single-path scenes. The former is for only the acrylic plate, and the latter is for only the mirror. Figure 13c is for the dual-path situation. Two reflections from both acrylic plate and mirror are overlapped. Note that the interference light from the acrylic plate illuminated the bottom half of the image sensor. Figure 14a,b show the reproduced depth images (33 × 52 pixels) for the acrylic plate placed at 1 m and the mirror placed at 9.3 m, respectively. The whole image sensor was illuminated. Figure 14c,d shows the depths reproduced from the images in Figure 13 for the dual-path scenario.

The total processing time for the dual-path scene was 74 min. The percentages of the processing time for the first stage (coarse depth estimation by TVAL3) and the second stage (depth refinement by fminsearchbnd) were 0.7% and 99.3%, respectively. Specifications of the PC are as follows: CPU: Intel Core i7-9700 (3 GHz, eight cores), RAM: 16 GB. Signal processing was conducted on MATLAB Version 9.3.0.713570.

Figure 15 shows the depth histograms for the single-path and dual-path situations. Table 4 summarizes the mean peak positions of the histograms, where the pixels with pixel values more than half of the maximum pixel value are considered. As shown in the table, multiple depth components were separated with an absolute error less than 0.2 m compared with the depths acquired in the single-path scenario.

## 5. Limitations

In the simulation, one of the drawbacks of the proposed method is a relatively large depth error when the two light reflections are very close. In Figure 8, the depth error of the interference light is significantly large between 14 m and 18 m. This can happen when the two pulses are merged. The most effective way to improve the separation is to speed up the operating frequency of the image sensor to shorten the minimal time window duration. Because the sensor is still a prototype, it has problems such as low photosensitivity, large pixel size, and slow operating frequency.

In Figure 9e–h, a huge error is observed at 26 m. At this distance, $d_{1}$ and $d_{2}$ are estimated to be 17.96 m and 24.39 m when no shot noise is considered, although they should be 16 m and 26 m, respectively. Figure 16 compares the simulated sensor output $p^{\prime }$ for $d_{1}$ = 16 m, $d_{2}$ = 26 m, and $d_{1}$ = 17.96 m, $d_{2}$ = 24.39 m. The signals are similar in terms of the peak positions. Therefore, the solution could converge to a local minimum. A depth refinement method or better shutter patterns that can find the global minimum should be explored.

The number of resolvable paths is of great concern. Basically, multi-path resolution using the shutter patterns shown in Figure 5 is a combinatorial problem. We assume a situation that the reflected light from an object placed at depth $d_{i}$ is detected by our computational CMOS image sensor using n-tap charge modulators. For the shutter patterns for a shutter period of $T_{j}$, $k_{j}$-th time window detects the reflection. $k_{j}$ is denoted by
(15)kj=⌊2dic mod Tjn⌋+1.

Here, ⌊x⌋ means the integer that is not more than x. In our proposed method, we prepare multiple $T_{j}$’s. Note that sub-clock shift is not considered here. For different $T_{j}$’s, $k_{j}$’s are detected from the captured image. However, when there are multiple reflections, $k_{j}$ becomes ambiguous. Namely, it is not easy to tell which signal for each $T_{j}$ comes from the same reflection. When signal peaks are separated for a certain $T_{j}$ and $k_{j}$ is specified for a reflection, we can resolve the multi-path interference more easily. There are two cases: (1) $k_{j1}$ (for higher frequency) is resolved, but $k_{j2}$ is not determined. (2) $k_{j2}$ (for lower frequency) is resolved, but $k_{j1}$ is not determined. For case 1, $k_{j2}$ is written as follows when $T_{j2}=2T_{j1}$:(16)kj2=⌊qTj1+(kj1−1)Tj1n mod Tj2n⌋+1.

Here, q is zero or a positive integer and mod means the modulation operator. The problem is to find q.

In case 2, another ambiguity arises. Because the time window duration for $T_{j2}$ is twice longer than that for $T_{j1}$, there are two choices as follows:(17)kj1=⌊(kj2−1)Tj2n mod Tj1n⌋+1 or⌊(kj2−1)Tj2n mod Tj1n⌋+2.

It is also necessary for the precision and accuracy to be small enough to determine the combinations of the detected signals for the multiple frequencies. To improve the capability of multi-path resolution, it is effective to increase the number of taps of the charge modulator. However, trade-off among the pixel size, response time, and separability should be considered.

Although only surface reflection was incorporated in the mathematical model in this paper, we believe that the proposed method could be extended to compensate multiple reflections and volumetric scattering because they significantly increase the path length or time of flight of light. The most difficult point is that the reflected light waveform is not the same as that of the emitted light. Probably, spatio-temporal point spread functions for these multi-path components should be also estimated. On the other hand, separation of sub-surface scattering could be more challenging and requires different techniques other than the application of temporal shutter patterns because the light intensity caused by the sub-surface scattering decays very quickly, e.g., in less than hundreds of picoseconds.

Improvement of the algorithm is also necessary. Although it is assumed that the incident light signal is uniform over a macro-pixel, we should consider the point spread function to realize subpixel resolution. Development of a non-iterative real-time signal processing algorithm is also key. In this paper, our aim was to investigate the potential of the multi-tap macro-pixel computational CMOS image sensor in multi-path separation. As shown in Section 4.2, the processing time for the depth refinement stage occupies 99.3% of the total processing time. Most of the processing time was dedicated to the nonlinear optimization. As the next step, algorithms suitable for real-time processing should be explored [30].

## 6. Conclusions

In this paper, the separation of multi-path components in time-of-flight depth imaging using a multi-tap macro-pixel computational CMOS image sensor was demonstrated. The macro-pixel has 2 × 2 subpixels, and each subpixel is implemented by a four-tap lateral electric field charge modulator (LEFM). Thus, 16 images for different temporal shutters are acquired in a single shot. Multiple frequency–time windows and sub-clock shifting were adopted to separate the multi-path interference components and to achieve both long distance range and high distance resolution. The following two-step depth estimation was used to investigate the potential of the proposed scheme for multi-path component separation. In the simulation without depth-dependent decay of the number of photons, the absolute value of the relative mean error was <1.2% for both $d_{1}$ (objective depth) and $d_{2}$ (interference depth), except when $d_{2}$ was very small (at 1 m) or two reflections were merged (at 16 m). When the amplitudes of the two reflections were the same, the error at $d_{2}=26$ m became large, probably because the solution could converge to a local minimum. For the simulation considering the depth-dependent decay of light intensity, the nonlinearity of the interference light increased significantly at long distances, probably under the effect of the strong objective light. In the experiment, we separated two reflections from an acrylic plate placed at 1 m or 2 m and a mirror placed at 9.3 m. The absolute value of the mean error compared with the single-path situation was less than 0.2 m.

## Figures and Tables

**Figure 1 sensors-22-02442-f001:**
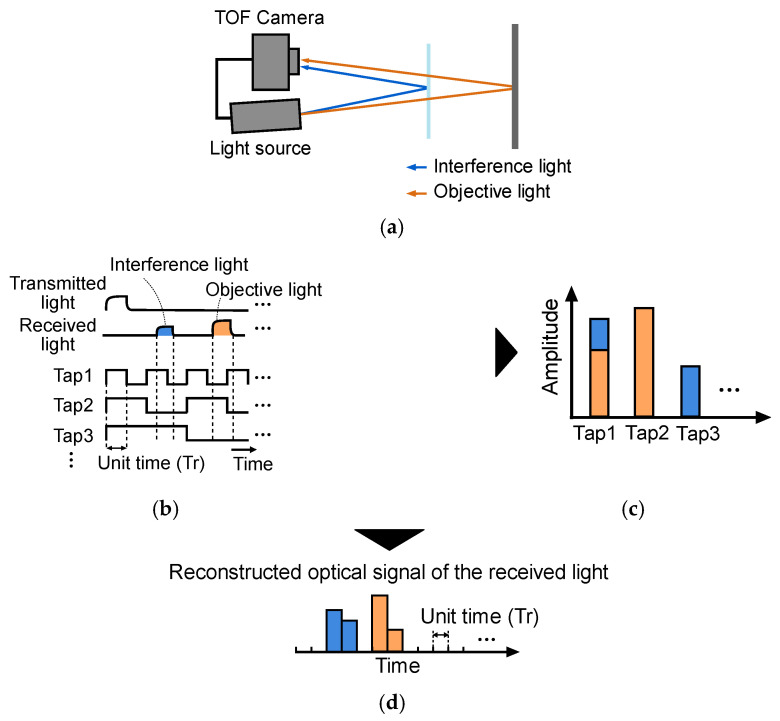
Separation of multi-path interference components: (**a**) optical setup that includes a transparent object causing multi-path interference, (**b**) compression of optical signals with temporal shutters, (**c**) compressed signals, and (**d**) reconstructed optical signal.

**Figure 2 sensors-22-02442-f002:**
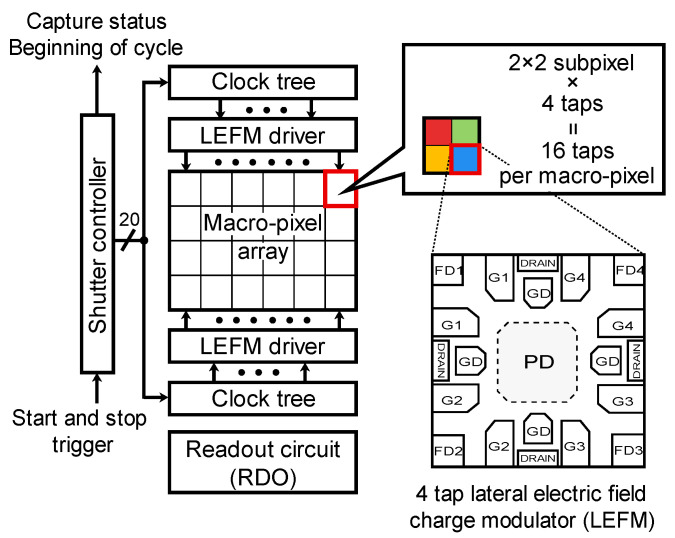
Structure of multi-tap macro-pixel computational CMOS image sensor. Each macro-pixel is composed of four subpixels. The subpixel is implemented by a four-tap LEFM with charge drain.

**Figure 3 sensors-22-02442-f003:**
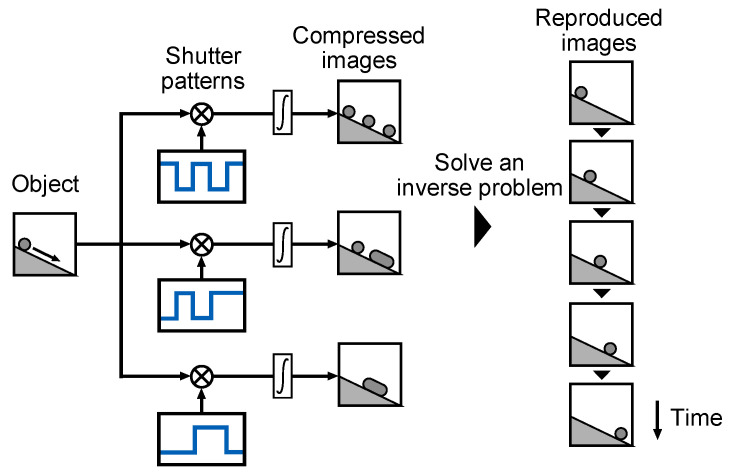
Image acquisition and reproduction flow of temporally compressive imaging. The object images are compressed in time with the temporal shutter patterns. Then, sequential images are reproduced by solving an inverse problem based on sparsity constraint.

**Figure 4 sensors-22-02442-f004:**
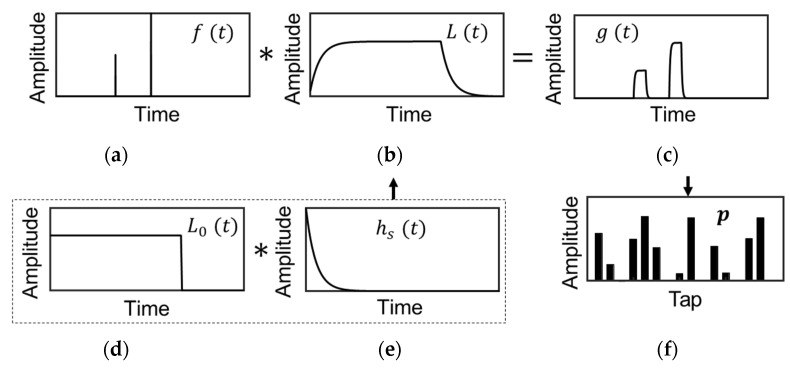
Modeling of multi-path interference in temporally compressive pulse iToF: (**a**) Scene response f(t). (**b**) Total system response L(t). (**c**) Detected light waveform g(t). (**d**) Light source waveform $L_{0}\left(t\right)$. (**e**) Sensor response $h_{s}\left(t\right)$. (**f**) Sensor output p.

**Figure 5 sensors-22-02442-f005:**
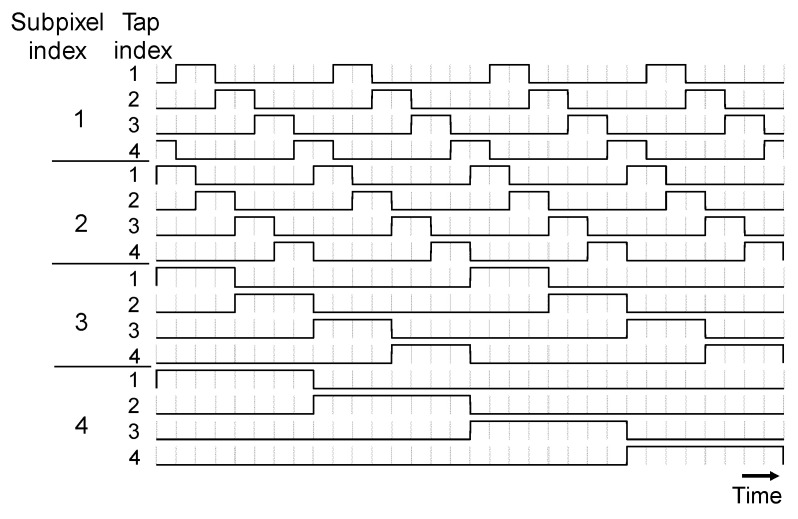
Temporally coded shutters (shutter length of 32 bits) with multiple frequencies and sub-clock shifting. The shutters of subpixel 1 are a half-clock-shifted version of those of subpixel 2.

**Figure 6 sensors-22-02442-f006:**
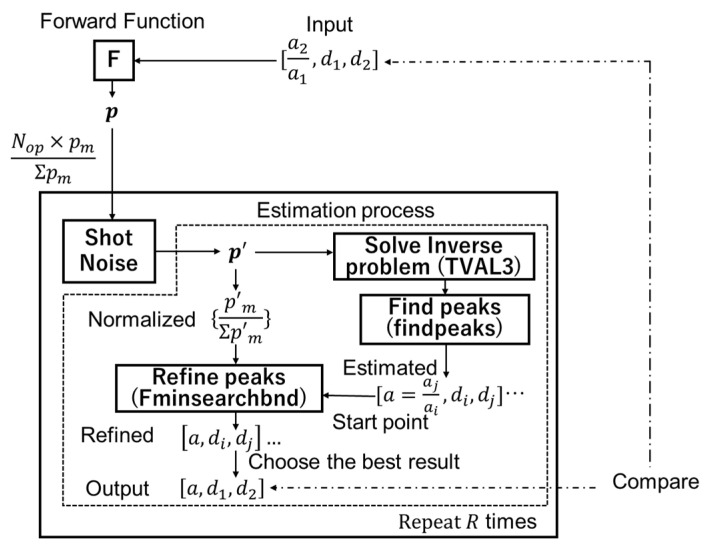
Simulation flow. The forward function generates a pixel value vector p from the depths of the objective and interference light and the amplitude ratio. These parameters are estimated in two stages from a noisy pixel value vector $p^{\prime }$ with shot noise.

**Figure 7 sensors-22-02442-f007:**
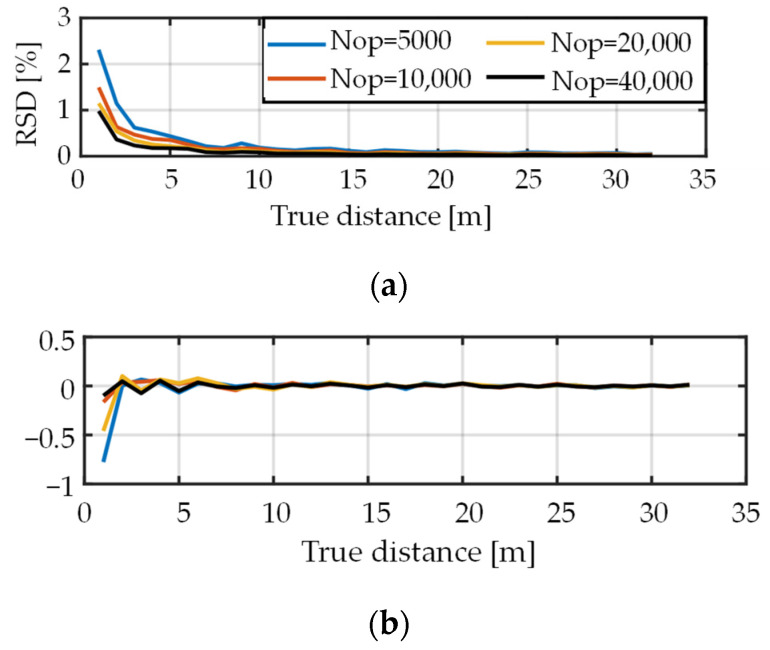
Results of single path simulation when depth-dependent decay of the number of photons is not considered. (**a**) Relative standard deviation and (**b**) relative mean error of the estimated depth.

**Figure 8 sensors-22-02442-f008:**
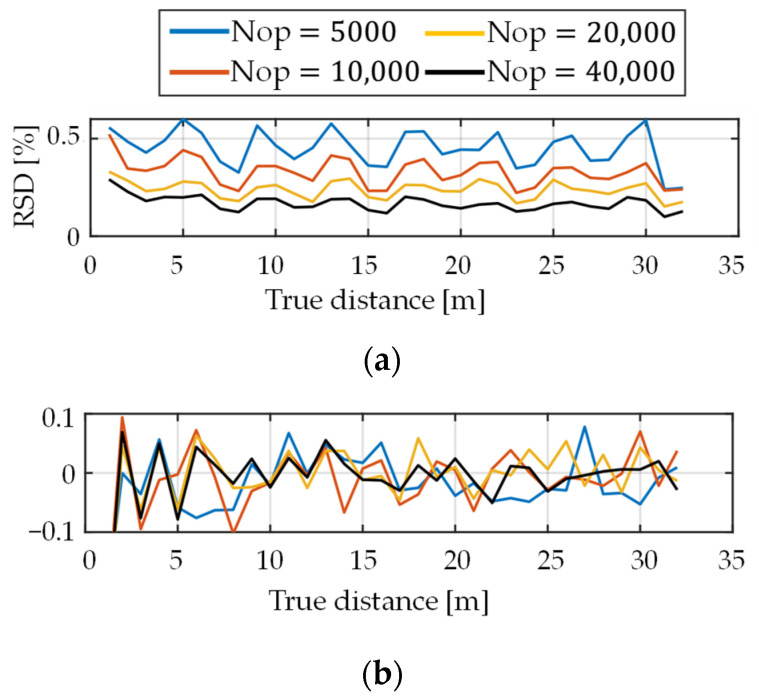
Results of single path simulation when depth-dependent decay of the number of photons is considered. (**a**) Relative standard deviation and (**b**) relative mean error of the estimated depth.

**Figure 9 sensors-22-02442-f009:**
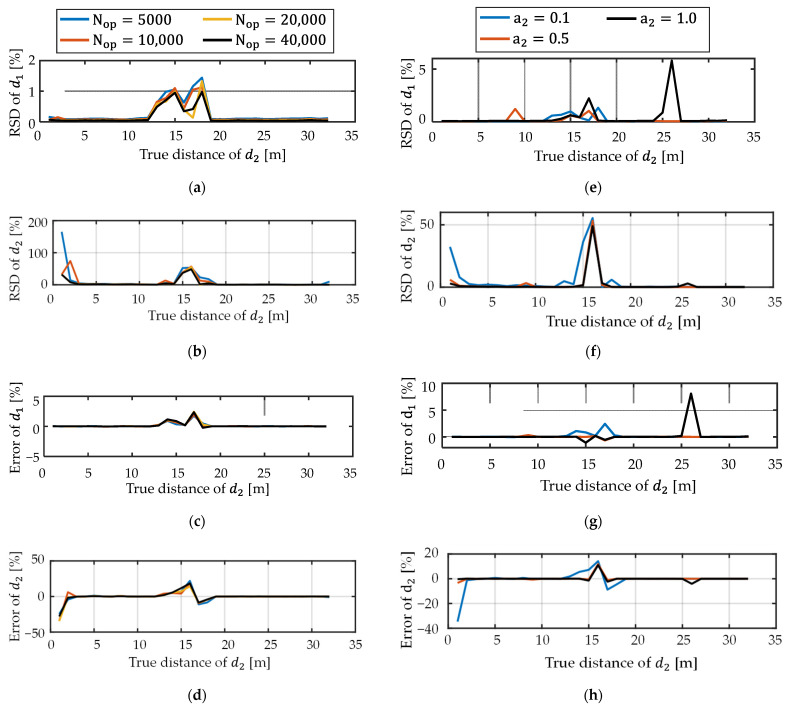
Results of dual-path simulations: (**a**–**d**) Type A. $N_{\mathrm{op}}$ was changed. (**e**–**h**) Type B. $a_{2}$ was changed. (**a**,**e**) Relative standard deviation of $d_{1}$. (**b**,**f**) Relative standard deviation of $d_{2}$. (**c**,**g**) Relative mean error of $d_{1}$. (**d**,**h**) Relative mean error of $d_{2}$.

**Figure 10 sensors-22-02442-f010:**
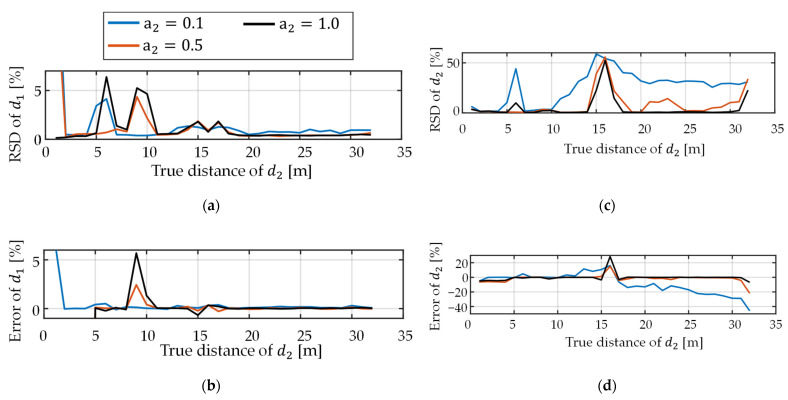
Results of dual-path simulations when depth-dependent decay of the number of photons was considered in Type B. $N_{\mathrm{op}}$ was set to 5000 at 4 m for the amplitude of 1. (**a**) Relative standard error and (**b**) relative mean error of $d_{1}$. (**c**) Relative standard error and (**d**) relative mean error of $d_{2}$.

**Figure 11 sensors-22-02442-f011:**
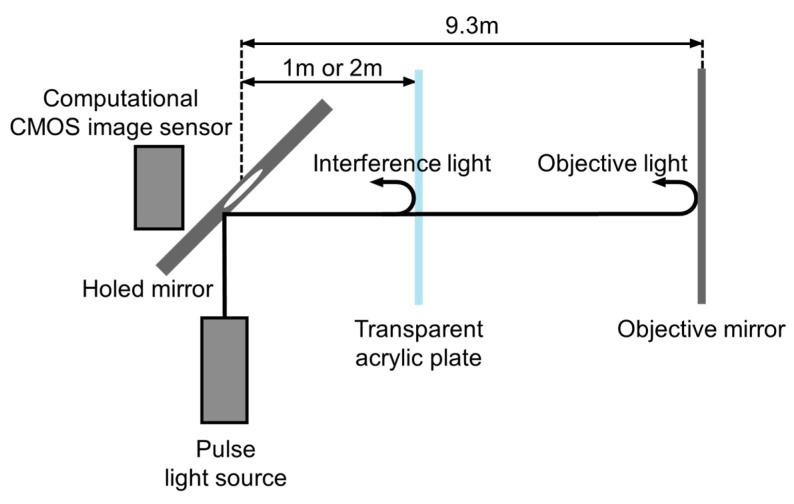
Experimental setup for dual-path ToF depth imaging. The objective mirror and transparent acrylic plate constitute dual-path interference.

**Figure 12 sensors-22-02442-f012:**
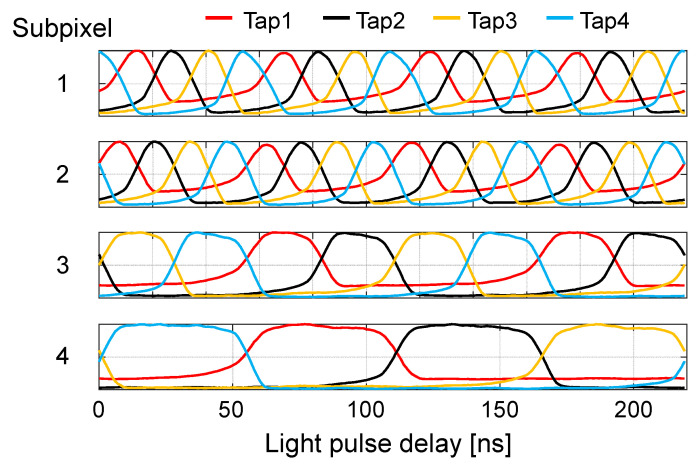
Measured sensor output versus light pulse delay. The waveforms are normalized by the maximum for each tap.

**Figure 13 sensors-22-02442-f013:**
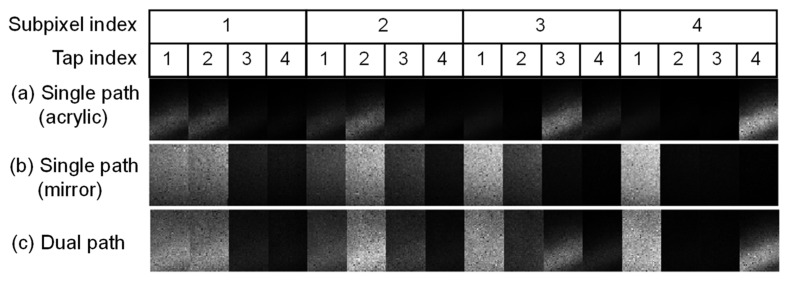
Captured images in the dual-path situation for the objective mirror and the acrylic plate placed at 9.3 m and 1 m, respectively. One hundred images are averaged.

**Figure 14 sensors-22-02442-f014:**
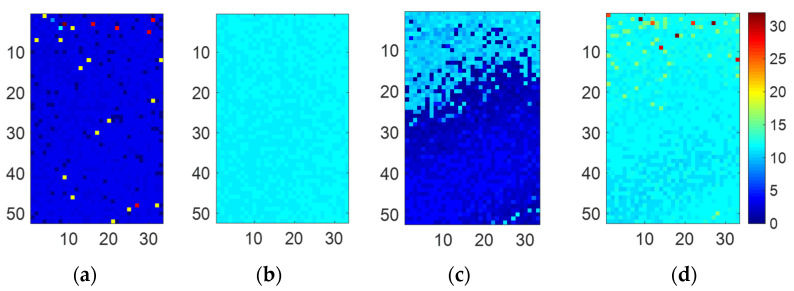
Measured depth images (33 × 52 pixels): (**a**) Acrylic plate placed at 1 m (single path). (**b**) Mirror placed at 9.3 m (single path). (**c**) Acrylic plate depth (dual-path). (**d**) Mirror depth (dual-path). In the dual-path situation (**c**,**d**), the depths are similar to those in the single-path situations (**a**,**b**).

**Figure 15 sensors-22-02442-f015:**
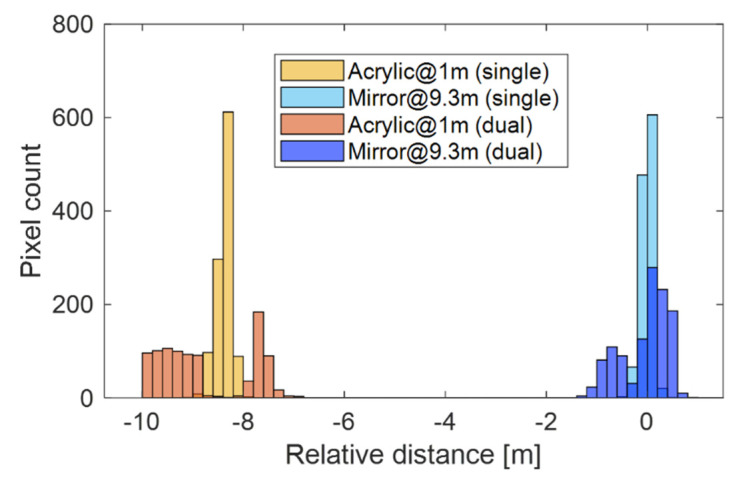
Depth histograms for the single-path and dual-path situations. The distance between the acrylic plate and the objective mirror was 8.3 m.

**Figure 16 sensors-22-02442-f016:**
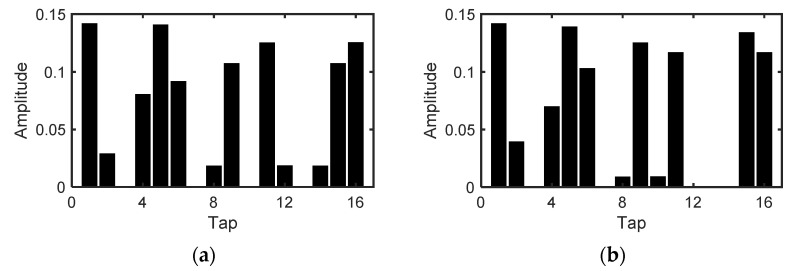
Simulated sensor output, $p^{\prime }$ without shot noise: (**a**) $d_{1}$ = 16 m, $d_{2}$ = 26 m. (**b**) $d_{1}$ = 17.96 m, $d_{2}$ = 24.39 m. They look very similar.

**Table 1 sensors-22-02442-t001:** Comparison of dToF and iToF.

	dToF	iToF
Detector	SPAD	Charge modulator
Pixel size	Relatively large	Small
Pixel readout circuits	Time-to-digital converter and histogram builder	The same as ordinary CMOS image sensors (pixel source followers, column correlated double sampling circuits, and analog-to-digital converters)
Immunity to multi-path interference	Good	No

**Table 2 sensors-22-02442-t002:** Specifications of multi-tap macro-pixel computational CMOS image sensor.

Technology	0.11 µm CMOS image sensor process
Chip size	7.0 mm × 9.3 mm
Valid subpixels	134 × 110
Subpixel pitch	22.4 µm × 22.4 µm
Subpixel count per macro-pixel	2 × 2
Tap count per subpixel	4
Shutter length per tap	8 to 256 bits by 8 bits

**Table 3 sensors-22-02442-t003:** Conditions of dual-path simulation.

		Type A (Variable Total Photons)	Type B (Variable Interference Reflection Amplitude)
Shutter length		32 bits	
Minimal time window duration		13.7 ns	
Light source pulse width		13.7 ns	
Number of total taps per macro-pixel		16	
Number of total electrons (*N*_op_)		5000, 10,000, 20,000, 40,000	20,000
Amplitude	Objective	$a_{1}=1$	$a_{1}=1$
	Interference	$a_{2}=0.1$	$a_{2}=0.1,\;0.5,\;1.0$
Depth	Objective	$d_{1}=16\;m$	$d_{1}=16\;m$
	Interference	$d_{2}=1-32\;m$	$d_{2}=1-32\;m$

**Table 4 sensors-22-02442-t004:** Measured relative mean depths based on the single-path situation (m).

	Acrylic Plate at 1 m	Acrylic Plate at 2 m	Mirror at 9.3 m
Single path	−8.5172	−7.2929	0
Dual path	−8.3404	−7.3693	−0.1826

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
