# Peer review of "Resolving Multi-Path Interference in Compressive Time-of-Flight Depth Imaging with a Multi-Tap Macro-Pixel Computational CMOS Image Sensor"

_sensors, 2022, doi:10.3390/s22072442_

Round 1

Reviewer 1 Report

This paper explores a new method to resolve a multi-path interference problem in iToF sensors. The proposed solution utilizing a multi-tap iToF pixel structure and a compressive sensing technique is interesting and valuable to readers of Sensors journal, although the manuscript has several points to be amended before publication. My questions and comments are as follows.

1. The compressive sensing with a multi-tap pixel looks quite complicated in solving inverse problems. I'm wondering about computational power and power consumption. Please comment on it.

2. The compressive ToF imaging in Section II is not easy to follow. Could you please define the shutter length and l_0-norm?

3. The relative standard deviation (RSD) and relative error also confuse me. Are they the same meaning as the precision (or uncertainty) and the accuracy (or nonlinearity) in the conventional iToF performance? Please clarify them in detail.

4. I'm curious about the simulation results in Fig. 7 and 8. The number of photons is identical regardless of the distance in the simulation, which is not realistic. It would be good to consider the number of photons with respect to the distance.

5. The RSD and relative error become large as a2 increases in Fig. 8, but it is described oppositely in a 250-line. Please correct it.

6. I'm wondering how many paths can be resolved by the proposed method. Could you please comment on the extendability?

7. The multi-path can be resolved by the multi-gate multi-tap iToF pixel operation presented by the same group, A Time-of-Flight Range Sensor Using Four-Tap Lock-In Pixels with High near Infrared Sensitivity for LiDAR Applications, Sensors, 2020, 20(1), 116. It also samples several depths in a pixel using multiple taps, resolving the multi-path problem without complicated processing. Please compare it with the proposed compressive method and describe the pros and cons.

Author Response

Thank you so much for your valuable suggestions.
I have tried to reflect your comments on the revision as much as possible.

>  1. The compressive sensing with a multi-tap pixel looks quite complicated in solving inverse problems. I'm wondering about computational power and power consumption. Please comment on it.

It is very difficult to estimate the power.
Therefore, measured processing time and the specifications of PC are added in Sec. 4.2.

>  2. The compressive ToF imaging in Section II is not easy to follow. Could you please define the shutter length and l_0-norm?

Definitions have been added in Sec. 2.1.

> 3. The relative standard deviation (RSD) and relative error also confuse me. Are they the same meaning as the precision (or uncertainty) and the accuracy (or nonlinearity) in the conventional iToF performance? Please clarify them in detail.

I am sorry for the confusing expressions. Yes, you are right.
In Sec. 3.1, the meanings are added the first paragraph of Sec. 3.1.

> 4. I'm curious about the simulation results in Fig. 7 and 8. The number of photons is identical regardless of the distance in the simulation, which is not realistic. It would be good to consider the number of photons with respect to the distance.

This is a very important point. I have added an explanation of why the number of photons is not dependent on the depth and simulations considering the depth-dependent decay of the number of photons. Please see new Figures 8 and 10.

> 5. The RSD and relative error become large as a2 increases in Fig. 8, but it is described oppositely in a 250-line. Please correct it.

I am sorry that the plots and explanations are confusing. I have modified the explanations of the axes for clarification. The explanation in Sec. 3.3 in the 2nd paragraph has been also corrected.

-> RSD of   d1 (objective reflection) became large as  Nop decreased or a2 increased especially when the objective light and interference light were merged. 

> 6. I'm wondering how many paths can be resolved by the proposed method. Could you please comment on the extendability?

The discussion has been added in Sec. 5.

> 7. The multi-path can be resolved by the multi-gate multi-tap iToF pixel operation presented by the same group, A Time-of-Flight Range Sensor Using Four-Tap Lock-In Pixels with High near Infrared Sensitivity for LiDAR Applications, Sensors, 2020, 20(1), 116. It also samples several depths in a pixel using multiple taps, resolving the multi-path problem without complicated processing. Please compare it with the proposed compressive method and describe the pros and cons.

My opinion has been added in Sec. 2.3.

I hope that the revision fulfills your requirements.

Reviewer 2 Report

The paper presents a very interesting and original multi-tap macro-pixel computational CMOS image sensor that is solving the problem of multi path interference caused by surface reflections. The proposed macro-pixel is composed of four subpixels embodied by a four-tap lateral electric field modulator. Simulations and experiments regarding the proposed method were provided.

The paper is well written, clearly, and well organized. The scientific is high and well sustained by a good mathematical support.

I remark the introduction and chapter 2 where the indirect time-of-flight based sensors are presented. The presentation is based on many references and in the same time on the previous research made by the authors.

I also remark the results of simulations and those obtained in the experimental part.

Question: Is the proposed sensor suitable other types of multi-path interference, other than of those given by the surface reflections?

The paper is very interesting, the results are convincing and strong argued and that’s why I consider that it may be accepted for publication.

Author Response

Thank you so much for reviewing my paper and for your valuable suggestion.

> Question: Is the proposed sensor suitable other types of multi-path interference, other than of those given by the surface reflections?

Although I have not tried to resolve the other multipath interference,
my speculation has been added in Sec. 5.

Thank you again.

Reviewer 3 Report

No comments an notes

Author Response

Thank you for reviewing!

Reviewer 4 Report

In this paper, a multi-tap macro-pixel computational CMOS image sensor is proposed to provide a solution for solving multipath interference caused by surface reflection. In depth estimation, sub clock refinement is introduced to facilitate the separation of multipath components, which is a certain degree of innovation. The purpose of this study is to investigate the potential performance of the proposed scheme. The author not only provides sufficient theoretical support, but also verifies the feasibility of this method by modeling and simulation, which is further supported by experiment of a multi-tap macro-pixel computational CMOS image sensor. The design is reasonable because it focuses on exploring the feasibility of the scheme and the potential of the sensor. However, the relationship between imaging results and conclusions is not well described.

Overall, the article is organized and its presentation is good. However, some minor issues still need to be improved. It is described in parts as follows:

In section 1, it provides sufficient technical background and briefly describes the current situation of solutions to multipath interference. The scheme refers to two cases in detail and explains the chapter structure. However, it is confusing when comparing itof and dtof. It is suggested that a table can be used for auxiliary explanation.

In section 2, the principle of the whole sensor and the mathematical model of the imaging process are mainly explained, which provides a theoretical basis for the simulation stage. The logic of this part is relatively strong and the expression is relatively clear. However, when describing the original input signal ?, the content is redundant.

In section 3, the main purpose is to provide simulation methods first, then to simulate the single and double paths, which is logically rigorous. However, in the single-path simulation section, there are few explanations of Figure 7, and there is no conclusive statement. In the two-path simulation section, the contents in Table 2 are not intuitive enough, and the explanations in the text are not clear enough. In addition, this section lacks information on the purpose of choosing single and double paths and comparing the two scenarios.

In section 4, the experiment is mainly carried out through the experimental device, which is more persuasive. In the experiment step, the image captured by the sensor is used to rebuild the depth image, then the histogram is obtained from the depth image. Finally, the depth value of the multipath component is calculated. The list shows its absolute error and verifies the logic rigor. However, in the image captured by the sensor, it is recommended to add the case of single path.

Finally, in section 4 and 5, the error analysis of the above single-path and double-path depth estimation results is made. This article objectively explains the potential and development space of this scheme, and has some reference value for readers in this field.

Overall, the structure of the article is clear, and the feasibility of the scheme is fully verified from In general, simulation and experiment, the structure of the article is clear, and the feasibility of the scheme is fully verified from theory, simulation and experiment. It is suggested that charts can be more intuitive and rich. For some simple charts, more detailed descriptions are needed.

About English expression:

There are still some statements that need to be modified appropriately. The recommendations are as follows:

1) The author frequently uses "with", in which the use of this word is not appropriate.

2) Commas are often used inappropriately, making some expressions prone to ambiguity.

3) The author often uses one sentence to express complex situations, which can make the semantics difficult to understand and sometimes even misinterpreted.

Author Response

Thank you so much for your valuable feedback.
I have modified the manuscript based on your comments.
Hope the revision is suitable for publication.

> However, it is confusing when comparing itof and dtof. It is suggested that a table can be used for auxiliary explanation.

I have added a comparison table as Table 1.

> However, when describing the original input signal ?, the content is redundant.

I must admit that the content is redundant. However, for clarification,
I would like you to accept the current form on this matter.

> However, in the single-path simulation section, there are few explanations of Figure 7, and there is no conclusive statement.

I have added the reason why the single-path scenario is considered in the 2nd paragraph of Sec. 3.1.
Some conclusion has been added in the 1st paragraph of Sec. 3.2.

> In the two-path simulation section, the contents in Table 2 are not intuitive enough, and the explanations in the text are not clear enough. 

I have added explanations in Table 3 (previously Table 2) for better understanding.

> In addition, this section lacks information on the purpose of choosing single and double paths and comparing the two scenarios.

The information has been added in the 2nd paragraph of Sec. 3.1.

> However, in the image captured by the sensor, it is recommended to add the case of single path.

The captured images for the single-path situations have been added in Figure 13 (previously Figure 11).

> It is suggested that charts can be more intuitive and rich. For some simple charts, more detailed descriptions are needed.

I have added more information to the captions that were too short.

> 1) The author frequently uses "with", in which the use of this word is not appropriate.

Inappropriate "with" was replaced by "of", "using", "by" , and so on.

>  2) Commas are often used inappropriately, making some expressions prone to ambiguity.

Unnecessary commas were removed.

> 3) The author often uses one sentence to express complex situations, which can make the semantics difficult to understand and sometimes even misinterpreted.

Lengthy sentences have been separated.

Thank you so much again for your contribution to my paper.